# On the Use of Gallic Acid as a Potential Natural Antioxidant and Ultraviolet Light Stabilizer in Cast-Extruded Bio-Based High-Density Polyethylene Films

**DOI:** 10.3390/polym12010031

**Published:** 2019-12-23

**Authors:** Luis Quiles-Carrillo, Sergi Montava-Jordà, Teodomiro Boronat, Chris Sammon, Rafael Balart, Sergio Torres-Giner

**Affiliations:** 1Technological Institute of Materials (ITM), Universitat Politècnica de València (UPV), Plaza Ferrándiz y Carbonell 1, 03801 Alcoy, Spain; sermonjo@mcm.upv.es (S.M.-J.); tboronat@dimm.upv.es (T.B.); rbalart@mcm.upv.es (R.B.); 2Materials and Engineering Research Institute, Sheffield Hallam University, Howard Street, Sheffield S1 1WB, UK; C.Sammon@shu.ac.uk; 3Novel Materials and Nanotechnology Group, Institute of Agrochemistry and Food Technology (IATA), Spanish National Research Council (CSIC), Calle Catedrático Agustín Escardino Benlloch 7, 46980 Paterna, Spain

**Keywords:** bio-HDPE, GA, natural additives, thermal resistance, UV stability, food packaging

## Abstract

This study originally explores the use of gallic acid (GA) as a natural additive in bio-based high-density polyethylene (bio-HDPE) formulations. Thus, bio-HDPE was first melt-compounded with two different loadings of GA, namely 0.3 and 0.8 parts per hundred resin (phr) of biopolymer, by twin-screw extrusion and thereafter shaped into films using a cast-roll machine. The resultant bio-HDPE films containing GA were characterized in terms of their mechanical, morphological, and thermal performance as well as ultraviolet (UV) light stability to evaluate their potential application in food packaging. The incorporation of 0.3 and 0.8 phr of GA reduced the mechanical ductility and crystallinity of bio-HDPE, but it positively contributed to delaying the onset oxidation temperature (OOT) by 36.5 °C and nearly 44 °C, respectively. Moreover, the oxidation induction time (OIT) of bio-HDPE, measured at 210 °C, was delayed for up to approximately 56 and 240 min, respectively. Furthermore, the UV light stability of the bio-HDPE films was remarkably improved, remaining stable for an exposure time of 10 h even at the lowest GA content. The addition of the natural antioxidant slightly induced a yellow color in the bio-HDPE films and it also reduced their transparency, although a high contact transparency level was maintained. This property can be desirable in some packaging materials for light protection, especially UV radiation, which causes lipid oxidation in food products. Therefore, GA can successfully improve the thermal resistance and UV light stability of green polyolefins and will potentially promote the use of natural additives for sustainable food packaging applications.

## 1. Introduction

The scarcity of petroleum and the great awareness about plastic waste have recently generated a great interest in the use of biopolymers for packaging applications [1]. Biopolymers include bio-based polymers, biodegradable polymers, and polymers featuring both characteristics. Bio-based polymers can successfully save fossil resources by using biomass that regenerates annually and provides the unique potential of carbon neutrality [2]. Bio-based polyethylene, also called “green” polyethylene, is a highly crystalline polyolefin produced by addition polymerization of ethylene obtained by catalytic dehydration of bioethanol [3]. Bio-based high-density polyethylene (bio-HDPE) has the same physical properties than its counterpart petrochemical resin, that is, high-density polyethylene (HDPE), showing good mechanical strength, high ductility, and improved water resistance [4,5]. In 2018, bio-based but non-biodegradable polyethylenes represented approximately 9.5% of the global bioplastics’ production capacity, reaching nearly 200,000 tons/year [6].

Polyolefins are excellent materials as the base of industrial plastic formulations due to their excellent balance between performance and processability by conventional processing routes such as extrusion and injection molding [7]. However, they are highly sensitive to degradation when exposed to oxidant atmospheres or ultraviolet (UV) light [8]. Polyethylene may undergo degradation, with subsequent increase in fragility, both during processing conditions by extrusion, that is, typically around 140–160 °C [9], or injection molding, that is, above 200 °C [10], and in the presence of light, heat, and chemicals. Hence, the addition of antioxidants and/or UV light stabilizers is habitually required to preserve its original physical properties for long periods. In this regard, phenolic compounds have been extensively used to extend the life service of low-density polyethylene (LDPE) [11,12]. Nevertheless, several synthetic polymer additives have been associated with toxicity effects on human health and the environment as well as other side effects such as carcinogenesis, which has led to some restraint in their use in plastics [13,14]. For instance, synthetic antioxidants such as polyphenol, organophosphate, and thioester compounds can potentially induce some toxicity derived from their migration into food products [15].

While scientific evidence on the exact implications is not conclusive, especially due to the difficulty of assessing complex long-term exposure, there are sufficient indications that warrant further research of natural additives for packaging manufacturers. For instance, tocopherol, plant extracts, and essential oils from herbs and spices have been proposed as natural antioxidants in polyolefins [16,17,18]. Other published works have reported the use of dihydromyricetin (DHM), quercetin or rosmarinic acid as UV light stabilizers [19,20]. Gallic acid (GA), that is, 3,4,5-trihydroxybenzoic acid, is a naturally occurring polyphenol commonly found in a variety of fruits and vegetables such as grapes, green tea, tea leaves or tomatoes [21,22]. Bioactive phenolic compounds can be effectively obtained by classical solid–liquid extraction employing organic solvents in heat-reflux systems [23] as well as other novel techniques including the use of supercritical fluids, high pressure processes, microwave-assisted extraction (MAE), and ultrasound-assisted extraction [24,25]. Therefore, GA is a good candidate to be applied as a natural polymer additive due to its natural origin, inherently low toxicity, and high bioactive activity such as antioxidant, anti-inflammatory, anticarcinogenic, and antifungal properties [26,27]. 

This study originally focuses on the use of the GA natural antioxidant to protect bio-HDPE from thermal and UV degradation. To this end, two contents of GA were melt-mixed during extrusion with bio-HDPE and the resultant materials were shaped into films by cast extrusion. The films were characterized in terms of their mechanical, morphological, and thermal performance as well as UV light stability to ascertain their potential in packaging applications.

## 2. Experimental

### 2.1. Materials

Bio-HDPE, SHA7260 grade, was manufactured by Braskem (São Paulo, Brazil) and supplied in pellet form by FKuR Kunststoff GmbH (Willich, Germany). This resin has a density of 0.955 g·cm^−3^ and a melt flow index (MFI) of 20 (2.16 kg, 190 °C). It has been developed for injection molding applications and its minimum bio-based content is 94%, determined by ASTM D6866. GA, with commercial reference G7384, having 97.5%–102.5% (titration) and 170.12 g·mol^−1^, was supplied in powder form by Sigma-Aldrich S.A. (Madrid, Spain). This is a water-soluble phenolic acid obtained from grapes and the leaves of different plants. 

### 2.2. Manufacturing of Films

Different mixtures of bio-HDPE and GA were manually premixed in a zipper bag and melt-compounded in a co-rotating twin-screw extruder from Construcciones Mecánicas Dupra, S.L. (Alicante, Spain). This extruder has a ratio of length (*L*) to diameter (*D*) ratio, that is, *L*/*D*, of 24, whereas its screws have a diameter of 25 mm. The speed of the screws was set at 20 rpm and the temperature profile was adjusted as follows: 145 °C (hopper)–150 °C–160 °C–165 °C (die). The extruded materials were cooled in air and then pelletized using an air-knife unit. GA was added at 0.3 and 0.8 parts per hundred resin (phr) of bio-HDPE, whereas a neat bio-HDPE sample was prepared in the same conditions as the control sample.

The compounded pellets were, thereafter, cast-extruded into films using a cast-roll machine MINI CAST 25 from EUR.EX.MA (Venegono, Italy). The extrusion speed was set at 25 rpm and the temperature profile was 150 °C (feeding)–155 °C–160 °C–165 °C–165 °C–170 °C–170 °C (head). Bio-HDPE films with an average thickness of approximately 150 μm were obtained by adjusting the speed of the calendar and the drag.

### 2.3. Color Measurements

A Hunter Mod. CFLX-DIF-2 colorimeter (Hunterlab, Murnau, Germany) was used to determine the color coordinates of the film samples. The values of *L** (lightness), *a** (red to green), and *b** (yellow to blue) parameters were determined while the color difference between two samples (Δ*E*_ab_*) was calculated using Equation (1):(1)$$\Delta E_{ab}{}^{*}=\sqrt{\Delta L^{*2}+\Delta a^{*2}+\Delta b^{*2}}$$
where Δ*L**, Δ*a**, and Δ*b** represent the differences in *L** and the *a** and *b** coordinates, respectively, between the neat bio-HDPE film and the GA-containing bio-HDPE films. At least five readings were taken for each film and the average values were reported. The following assessment was used to evaluate the color change of the films based on the Δ*E*_ab_* values: below 1 indicates an unnoticeable difference in color; 1–2 a slight difference that can only be noticed by an experienced observer; 2–3.5 a noticeable difference by an unexperienced observer; 3.5–5 a clear noticeable difference; and above 5, different colors are noticeable [28]. 

### 2.4. Mechanical Tests

A universal test machine Elib 50 from S.A.E. Ibertest (Madrid, Spain) was used to perform the tensile tests in the bio-HDPE film samples following the guidelines of ISO 527-1:2012. The selected load cell was 5 kN and the cross-head speed was set at 10 mm·min^−1^. Standard tensile samples (type 2) with a total length and width of 160 and 10 mm, respectively, were tested as indicated in ISO 527-3. Tests were performed at room conditions and at least six samples per film were analyzed.

### 2.5. Thermal Characterization

The main thermal transitions of the bio-HDPE film samples were obtained by differential scanning calorimetry (DSC) in a Mettler-Toledo 821 calorimeter (Mettler-Toledo, Schwerzenbach, Switzerland). Samples with a total weight of about 5–10 mg were placed into aluminum crucibles. Two types of DSC tests were carried out to evaluate the antioxidant efficiency of GA. The first test was based on a dynamic program from 30 to 350 °C in an air atmosphere at a heating rate of 5 °C·min^−1^ where the oxidative degradation was identified as the onset oxidation temperature (OOT). The second test consisted of a heating ramp from 30 to 210 °C in an air atmosphere at a heating rate of 5 °C·min^−1^, followed by an isotherm at 210 °C for a whole period of 400 min. The latter test allowed for the oxidation induction time (OIT) to be obtained. Furthermore, the degree of crystallinity (*X*_C_) was calculated following Equation (2):(2)$$X_{C}=\left[\frac{\Delta H_{m}-\Delta H_{CC}}{\Delta H_{m}^{0}\Delta \left(1-w\right)}\right]\cdot 100$$
where Δ*H*_m_ (J·g^−1^) and Δ*H*_CC_ (J·g^−1^) correspond to the melt and cold crystallization enthalpies, respectively. Δ*H*_m_^0^ (J·g^−1^) stands for the melt enthalpy of a theoretically fully crystalline of bio-HDPE with a value of 293.0 J·g^−1^ [29] and the term *1-w* represents the weight fraction of bio-HDPE.

Thermal stability was also determined by thermogravimetric analysis (TGA) in a Mettler-Toledo TGA/SDTA 851 thermobalance (Mettler-Toledo, Schwerzenbach, Switzerland). Samples with an average weight of 5–7 mg were placed in standard alumina crucibles (70 μL) and subjected to a heating program from 30 to 700 °C in air atmosphere at heating rates of 20 °C·min^−1^. All the thermal tests were performed in triplicate.

### 2.6. Aging Treatment

The aging treatment of materials was performed by means of a high-pressure mercury lamp, with 1000 W and 350 nm wavelength, model UVASPOT 1000RF2 (Honle Spain S.A., Barcelona, Spain) in a closed chamber under ambient conditions. Samples were exposed for a period of up to 10 h and tests were carried out in triplicate.

### 2.7. Infrared Spectroscopy

Attenuated total reflection–Fourier transform infrared (ATR-FTIR) spectroscopy was used to perform chemical analysis of the films. A Vector 22 from Bruker S.A. (Madrid, Spain) coupling a PIKE MIRacle™ ATR accessory from PIKE Technologies (Madison, WI, USA) was used to record the FTIR spectra. Ten scans were averaged from 4000 to 450 cm^−1^ at a resolution of 4 cm^−1^. Film samples that were UV treated at 30 min intervals were used to collect variable time FTIR spectra for a whole span time of 10 h.

### 2.8. Microscopy

The morphology of the fracture surfaces of the UV-treated films of bio-HDPE was observed by field emission scanning electron microscopy (FESEM) in a ZEISS ULTRA 55 from Oxford Instruments (Abingdon, UK). Samples were obtained by cryo-fracture and an acceleration voltage of 2 kV was applied during FESEM observation. The surfaces were previously coated with a gold-palladium alloy in an EMITECH sputter coating SC7620 model from Quorum Technologies, Ltd. (East Sussex, UK).

## 3. Results and Discussion

### 3.1. Optical Properties of the GA-Containing Bio-HDPE Films

Figure 1 shows the surface view of the bio-HDPE films varying the GA content. Simple naked eye examination of these images indicated that all of the biopolymer films showed a high contact transparency. Indeed, bio-HDPE is highly transparent due to its high crystalline nature [30]. All the film samples exhibited a smooth, defect-free, and uniform surface, in which GA yielded a yellow color and also certain opacity. The latter effect can be ascribed to the presence of the GA particles, which reduced the transparency properties by blocking the passage of ultraviolet–visible (UV–Vis) light and scattering light. A similar yellowing effect was observed by Al-Malaica et al. [31], who reported the effect of changing the concentration of tocopherol and Irganox 1010 (a commercial phenolic antioxidant) on the color stability of polypropylene (PP). At low additive concentrations, both antioxidants showed low influence on the color sample, expressed in terms of differences in yellow index, whereas higher concentrations of tocopherol led to noticeable color changes. In order to quantify the optical parameters, Table 1 gathers the values of *L**, *a**, and *b** of all the bio-HDPE films and also the Δ*E*_ab_* values of the bio-HDPE films containing GA. One can observe that as the GA content increased, the luminance of the film decreased, confirming that the bio-HDPE films became less transparent. It could also be observed that the *a** coordinate slightly changed from negative values (green) to nearly neutral values, while the *b** coordinate also changed remarkably from negative values (blue) to positive values (yellow) [32]. Therefore, the incorporation of the here-tested GA loadings induced an increase in both opacity and the hue of yellow color, which could restrict the use of biopolymer films for transparent applications. Furthermore, the development of a different color in the bio-HDPE film after the GA addition was noticeable (Δ*E*_ab_* ≥ 5). However, the GA-containing bio-HDPE films can also offer some advantages for certain packaging applications. For instance, this optical property can be desirable for the protection of foodstuff from light, especially UV radiation, which can cause lipid oxidation in food products [33,34]. Examples include snack products that are made with refined vegetable oils and dried soups such as chicken soup that are sensitive to UV light because they contain highly sensitive unsaturated fatty acids or dry broccoli cream soup that is sensitive to visible light because it contains the photosensitizers chlorophyll from broccoli and riboflavin from dairy ingredients. Another potential application of the here-developed films is to avoid the discoloration of sliced sausage, which is a well-known adverse effect of light that often occurs even if the product is packed under vacuum [35].

### 3.2. Mechanical Properties of the GA-Containing Bio-HDPE Films

Tensile tests were carried out in order to analyze the mechanical properties of the GA-containing bio-HDPE films. Table 2 summarizes the values of tensile modulus (*E*_tensile_), maximum tensile strength (σ_max_), and elongation at break (ε_b_). One can observe that *E*_tensile_ of the neat bio-HDPE film was 292.5 MPa and this value was reduced to 222.1 and 243.6 MPa with the incorporation of 0.3 and 0.8 phr of GA, respectively. The value of σ_max_ was in the 20−21 MPa range for all of the bio-HDPE film samples, which is similar to the values reported by other authors [36]. In relation to ε_b_, the neat bio-HDPE film showed a value of 45.2%, which was also reduced to 18.6% and 20.2% after the incorporation of 0.3 phr and 0.8 phr of GA, respectively. Significantly higher ε_b_ values, around 450%–550%, have been reported for injection-molded articles of bio-HDPE [5,37], which can be ascribed to the testing conditions, processing method, and differences in the percentage of crystallinity as well as crystal orientation during manufacturing. Therefore, the incorporation of GA resulted in a reduction in both elasticity and ductility of bio-HDPE. In this regard, crystallinity can play a significant role in the mechanical and durability performance in rigid applications. A decrease in the polymer’s crystallinity can lead to a reduction of *E*_tensile_ and σ_max_, which are parameters ascribed to mechanical strength [38]. For instance, Jamshidian et al. [39] showed that the use of different antioxidants in polylactide (PLA) films yielded lower values of *E*_tensile_, σ_max_, and ε_b_. Whereas the reduction in mechanical strength was related to an effect of reduced crystallinity, the ductility impairment observed was ascribed to a phenomenon of stress concentration by the presence of additives with a low interfacial adhesion with the biopolymer matrix. Similarly, work performed by Jamshidian et al. [40] demonstrated that the addition of antioxidants in PLA films yielded a reduction in their mechanical performance due to the additive not being homogenously distributed throughout the entire polymer structure, which could lead to polymer inconstancy and be another reason for decreased mechanical parameters. In general, the incorporation of antioxidants and other polymer additives can alter the film continuity and then decrease the movement of the polymer chains, leading to a ductility decrease [41]. In the particular case of HDPE, its mechanical performance reduction has been attributed to the direct reaction of the antioxidant with oxygen that lowers the efficiency of the inhibitor, pro-oxidant transformation products that may be formed during the processing operations and can participate in oxidative degradation and, more importantly, to limitations in the solubility of antioxidants in the polyolefin matrix [42]. This effect differs from that of UV stabilizers, which tend to be more soluble in low-molecular weight (*M*_W_) organic solvents [43].

### 3.3. Thermal Properties of the GA-Containing Bio-HDPE Films

Both DSC and TGA tests were carried out in order to ascertain the influence of the GA addition on the thermal stability of the bio-based polyolefin. Figure 2 shows the dynamical DSC curves of the cast-extruded bio-HDPE films, whereas Table 3 summarizes the main thermal parameters obtained from the curves. One can observe that, in all cases, the polyolefin melted sharply in a single peak at approximately 132 °C. A similar melting profile has been observed previously for this polyolefin, regardless of the origin and the methodology followed to prepare the articles [37,44]. It can also be seen that the Δ*H*_m_ values of bio-HDPE slightly reduced as the GA content in the green polyolefin increased. In particular, the crystallinity degree, that is, *X*_C_, was slightly reduced from 54.8% for the neat bio-HDPE film to 53.5% and 52% for the bio-HDPE films containing 0.3 phr and 0.8 phr of GA, respectively. This result suggests that the presence of the GA antioxidant decreased the lamellae size of the bio-HDPE crystals by inducing imperfections [45]. For instance, Lopez-de-Dicastillo et al. [46] similarly reported that the incorporation of ascorbic acid, ferulic acid, quercetin, or green tea extract induced a lower and more deficient crystallinity structure for poly(ethylene–*co*–vinyl alcohol) (EVOH).

More interestingly, the DSC plots also revealed the significant oxidative retardant effect of GA on bio-HDPE. It can be observed that the onset of thermal degradation (*T*_onset_), also called OOT when the DSC run is carried out in an oxygen-rich environment, started at 226.3 °C in the neat bio-HDPE film. This value is relatively similar to that reported by Jorda-Vilaplana et al. [47], who showed that bio-HDPE started thermal degradation at approximately 232.5 °C. The value of *T*_onset_ then increased by 36.5 °C and nearly 44 °C in the bio-HDPE films containing 0.3 phr and 0.8 phr of GA, respectively. Similar results were obtained by Samper et al. [17] where 0.5 wt % silibinin and quercetin acted as oxidative retardants for PP as both natural additives successfully delayed the onset of thermal oxidation. In this sense, Dopico-Garcia et al. [48] showed that the use of natural antioxidants could successfully result in polyolefins with enhanced stabilization against thermal-oxidation degradation. The criteria for the antioxidant activity is based on the o-dihydroxy structure of their B-ring, which confers higher stability to the radical form and participates in electron delocalization for effective radical scavenging.

Figure 3 shows the isothermal DSC curves of the cast-extruded bio-HDPE films measured at 210 °C for a span time of 400 min. It can be observed that, after a heating ramp of 36 min, all the DSC pans reached 210 °C and the green polyolefin samples already melted and then showed similar curves in which, thereafter, oxidation occurred at different times. The OIT value, that is, the time between melting and the decomposition onset in isothermal conditions, was seen as an exothermic peak. One can notice that in the neat bio-HDPE film, oxidation initiated at approximately 5 min. The addition of 0.3 phr and 0.8 phr of GA successfully delayed oxidative thermal degradation of bio-HDPE by approximately 56 min and 240 min. It is worth noting that the high performance achieved herein using GA, a natural antioxidant, in comparison to other antioxidants. For instance, the use of other phenolic compounds such as the natural antioxidants naringin or silbinin at 0.5 wt % resulted in an OIT value of 17 min at 210 °C in PP [17]. In relation to synthetic antioxidants, Li et al. [49] showed that the incorporation of 0.1 wt % of dendritic antioxidant delayed the oxidation of PP and LDPE by 40 min and 50 min, respectively.

The improvement attained with the incorporation of the GA can also be related to the good dispersion of GA achieved within the bio-HDPE matrix. Thus, the additive chemically makes better contact with the peroxyradical on the polymer chains to inhibit the oxidation reaction. In this regard, Koontz et al. [50] showed that the addition of tocopherol improved the OIT value of linear low-density polyethylene (LLDPE) in 68 min when it was uniformly dispersed in the polyolefin matrix. Furthermore, the particular chemical structure of GA provides a great antioxidant capacity, which has been widely reported in food technology, medicine, pharmacy, etc. [51,52,53]. Consistent with most polyphenolic antioxidants, both the configuration and total number of hydroxyl groups can substantially influence its antioxidant activity mechanism [54,55]. In particular, GA is a free radical scavenger that is based on the high reactivity of the hydroxyl substituents (F−OH) that participate in the next reaction [56]:$$F-OH+R^{.}\to F-O^{.}+RH$$

Hence, when a free radical ($R^{.}$) is formed by thermo-oxidation, the phenolic compounds move toward this unstable point to block further degradation and produce a stabilization effect. Thus, F−OH donates hydrogen to become peroxyl ($F-O^{.}$), stabilizing the free radical. Among structurally homologous flavones and flavanones, peroxyl and hydroxyl scavenging increases linearly and curvy-linearly, respectively, according to the total number of hydroxyl groups [57].

Figure 4 shows the TGA curves (Figure 4a) and DTG curves (Figure 4b) of the cast-extruded bio-HDPE films while Table 4 summarizes the main thermal parameters obtained from the curves. The neat bio-based polyolefin presented an onset degradation temperature (*T*_onset_) of 256.9 °C. The temperature of maximum degradation (*T*_deg_), which corresponds to the temperature with the maximum degradation rate, was 427.8 °C. Although the thermal degradation of the green polyolefin was produced in a single step, a lower decomposition rate was observed up to approximately 370 °C, which can be seen as a shoulder in the DTG curve of the neat bio-HDPE. In this thermal range, the decomposition of the C–C covalent bond started and free radicals were generated. At higher temperatures, the free radicals formed led to sequential thermal degradation and breakdown of the main polyolefin chain [58]. Finally, all the film samples showed a similar residual mass of 0.2%–0.3%, indicating full thermal decomposition at 700 °C. In this sense, Montanes et al. [59] observed a similar thermal degradation profile for this green polyolefin, which was based on a one-step weight loss that ranged between 390 and 508 °C. The addition of 0.3 and 0.8 phr of GA successfully induced an improvement in the bio-based HDPE film of approximately 27 and 35 °C in the *T*_onset_ value, respectively, and suppressed the formation of the above-described free radicals. This thermal stability enhancement was relatively similar to that obtained above by DSC, as shown in previous Table 3, which is related to the intrinsic antioxidant activity of the natural polyphenol. In comparison to previous works using synthetic antioxidants, Zeinalov et al. [60] showed that thermal degradation of neat polystyrene (PS) started at around 270 °C, while the addition of 1% of antioxidant (Fullerene C60) delayed it up to 300 °C. In relation to other works reporting the use of GA, Luzi et al. [61] recently described that the addition of 5 wt % GA successfully increased the thermal stability of EVOH films by nearly 20 °C.

It is also worthy to note that the GA addition increased the values of *T*_deg_ by approximately 15 °C, but it also increased the of mass loss rate during thermal degradation. In general, the incorporation of different natural antioxidants can significantly improve the thermal stability of polymers. In particular, some authors have reported similar results with other natural antioxidants [62,63]. For instance, España et al. [64] showed that the incorporation of phenolic compounds successfully improved the *T*_deg_ values of green composites made of a mixture of lignin and organic coconut fibers (CFs) with an excellent stabilization provided by tannic acid.

### 3.4. Chemical Characterization of the GA-Containing Bio-HDPE Films

The chemical changes in the bio-HDPE films after the GA addition were analyzed by means of FTIR spectroscopy. Figure 5 shows the FTIR absorbance spectra of the GA, in powder form, and the neat bio-HDPE film and GA-containing bio-HDPE films. The main peaks of GA were observed at the 3100−3500 cm^−1^ region and, more intensely, from 1650 to 560 cm^−1^. The peak located at 3491 cm^−1^ is ascribed to the O–H stretching vibration of the hydroxyl groups of polyphenols [65]. Xu et al. [66] showed that the strong absorption peak at around 1614 cm^−1^ and bands between 1400 cm^−1^ and 1200 cm^−1^ are characteristic of polysaccharides. The peaks in the 1200–1000 cm^−1^ region originated from ring vibrations that overlapped with the stretching vibrations of C–OH side groups and the C–O–C glycosidic band vibration [67]. In relation to the green polyethylene, the intense peaks at 2925, 2850, 1460, and 725 cm^−1^ were respectively assigned to stretching vibrations and bending and rocking deformations of the methylene (CH_2_) groups [68]. Furthermore, the low-intense bands located between at 1377 and 1351 cm^−1^ were assignable to the wagging deformation and symmetric deformation of the CH_2_ and methyl (CH_3_) groups, respectively. The peaks in the 1700–1800 cm^−1^ and 1200–1300 cm^−1^ regions have been ascribed to carbonyl compounds formed in the oxidation products of polyethylene [69].

The incorporation of GA into the bio-HDPE film generated the appearance of a series of peaks, particularly noticeable for the film containing 0.8 phr, which confirmed the presence of the natural antioxidant in the polyolefin. Briefly, the hydroxyl groups of GA altered the bands related to the CH_2_ groups of bio-HDPE in the 3100−3500 cm^−1^ region. In addition, the formation of a new weak band at 1607 cm^−1^ can be ascribed to the stretching and bending vibrations of the aromatic ring of GA [65]. One can also observe the formation of a new low-intensity peak centered around 1030 cm^−1^. Bands formed between 1021 and 1037 cm^−1^ have been ascribed to the formation of dimers or oligomers of GA that can result from the stretching vibration of C−C and C−O bonds [65,70]. Indeed, GA is known to be auto-oxidized to its semiquinone free radicals, which can consequently generate hydroquinone [71].

### 3.5. UV Light Stability of the GA-Containing Bio-HDPE Films

The bio-HDPE films were subjected to UV light for a span time of up to 10 h, herein referred to as the aging time, in order to ascertain the influence of the GA addition on their UV light stability. Based on the above spectra, FTIR spectroscopy was used to analyze the chemical changes on the samples after being exposed to UV light. Figure 6 shows the 3D plots of the FTIR spectra taken across the exposure time to UV light. In the case of the neat bio-HDPE film, the UV exposure greatly increased the relative intensity of the strongest peaks observed at 2919 and 2851 cm^−1^, which are assigned to the CH_2_ antisymmetric and symmetric stretching, respectively [68]. Furthermore, the peaks centered at 1463 and 720 cm^−1^, which are respectively ascribed to bending and rocking deformations in polyethylene [68], also increased. These chemical changes suggested an increase of the CH_2_ groups in the film sample, which can be related to the partial breakdown of the polyolefin chain by UV light exposure and the formation of more terminal groups. It is also noteworthy that UV degradation was already noticeable after 30 min of UV light treatment, whereas it increased slightly in the whole aging time tested. No further changes were observed in the bands related to oxidized groups since changes were very subtle and are also known to appear after longer UV exposure periods and higher temperatures [72]. The fast degradation changes observed in bio-HDPE can be ascribed to the above-reported mechanism based on free radicals with high reactivity. Interestingly, these absorbance bands related to CH_2_ compounds of both GA-containing bio-HDPE films remained nearly constant and a slight increase was observed after 4 h of UV light exposure. Therefore, the addition of GA successfully improved the oxidation stability of the green polyolefin, also offering long-term UV stability. As also explained above during the thermal analysis, the chemical configuration of GA and the significant number of hydroxyl groups could successfully stabilize the free radicals formed during UV light exposure.

Finally, Figure 7 shows the FESEM images of the bio-HDPE films exposed to 5 h UV light. One can observe that, prior to UV light exposure, all of the films presented a similar fracture surface without any cracks or wrinkles. After 1 h of UV light exposure, the films developed an increase in roughness on their fracture surfaces. However, the GA-containing films generated a smoother surface, which is representative for a slower or negligible UV light aging. One can also observe that the fracture surface of the neat bio-HDPE suffered a remarkable modification after 2.5 h of UV light exposure. Indeed, the life time of an article made of HDPE without stabilizers can be as low as one year since the polyolefin decomposes rapidly by UV light action [73]. This phenomenon is related to the presence of impurities that are formed during their synthesis such as carbonyl, peroxide, hydroxyl, hydroxyperoxide, or any substances with unsaturated groups, which absorb light at higher wavelengths and thus yield the generation of free radicals. The incorporation of 0.3 phr GA significantly reduced the UV degradation of bio-HDPE and the fracture surfaces remained similar for up to 2.5 h, time at which some wrinkles were formed. Furthermore, 0.8 phr GA successfully kept the film samples stable up to 5 h of UV light exposure, showing fracture surfaces free of cracks. Therefore, the morphological analysis correlates well with the FTIR spectroscopy results shown above and confirmed the UV light stability provided by GA to the bio-HDPE films that can thus improve the shelf life of the green polyolefin. The antioxidant effect of GA is considered to also protect the free residues that are generated during polymer synthesis with improved degradation stability [74]. Indeed, UV stabilizers have always been categorized as a subgroup within the antioxidant additive group. Similarly, Du et al. [75] showed that HDPE/wood flour composites containing pigments presented fewer cracks on the surface than composites without pigment after accelerated UV weathering. The authors suggested that pigments can mask some UV radiation and prevent HDPE against UV radiation damage. Similar results were previously reported by Samper et al. [17] through the use of quercetin and silibinin as UV light stabilizers for PP.

## 4. Conclusions

This work describes the development and characterization of cast-extruded bio-HDPE films containing the natural antioxidant GA in order to ascertain their potential application in food packaging. The incorporation of GA at 0.3 and 0.8 phr contents induced a mechanical elasticity and ductility impairment and also a crystallinity reduction of bio-HDPE due to its limited solubility in the green polyolefin matrix. The bio-HDPE films also developed a low-intense yellow color but were still contact transparent. Interestingly, the OOT values was delayed by 36.5 °C and nearly 44 °C while the OIT values were reduced by approximately 56 and 240 min in the bio-HDPE films containing 0.3 and 0.8 phr GA, respectively. Furthermore, the UV light stability of bio-HDPE was significantly improved after the GA addition for an aging time monitored by FTIR spectroscopy of 10 h. The enhancement attained was ascribed to the high capacity of the phenolic compounds present in the natural antioxidant to stabilize the free radicals formed during degradation of the green polyolefin. As a result, GA can be regarded as a natural antioxidant and UV light stabilizer that can potentially replace synthetic additives in biopolymer formulations for food packaging applications following the Bioeconomy principles. Nevertheless, future studies should be addressed to increase the ductility of the resultant biopolymer films by, for instance, the addition of natural plasticizers, while the analysis of their barrier properties and the performance of specific migration tests will also be required according to the targeted application.

## Figures and Tables

**Figure 1 polymers-12-00031-f001:**

Visual appearance of the bio-based high-density polyethylene (bio-HDPE) films containing different amounts of gallic acid (GA): (**a**) Bio-HDPE; (**b**) Bio-HDPE + 0.3GA; (**c**) Bio-HDPE + 0.8GA.

**Figure 2 polymers-12-00031-f002:**
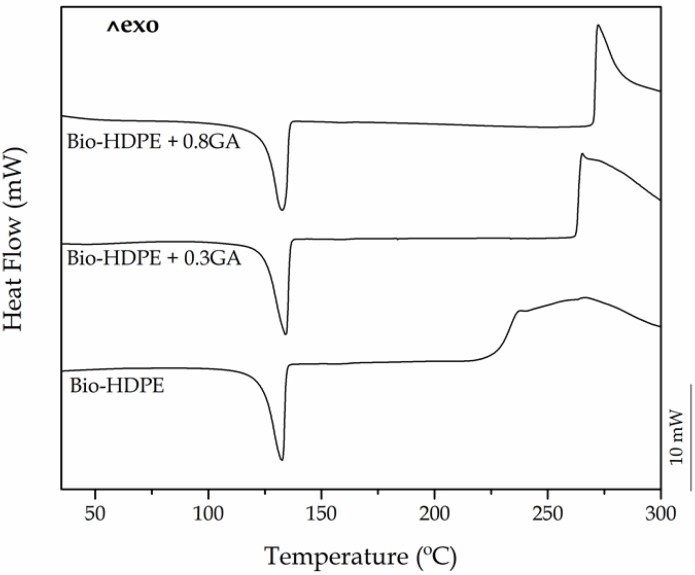
Differential scanning calorimetry (DSC) heating curves of the bio-based high-density polyethylene (bio-HDPE) films containing different amounts of gallic acid (GA).

**Figure 3 polymers-12-00031-f003:**
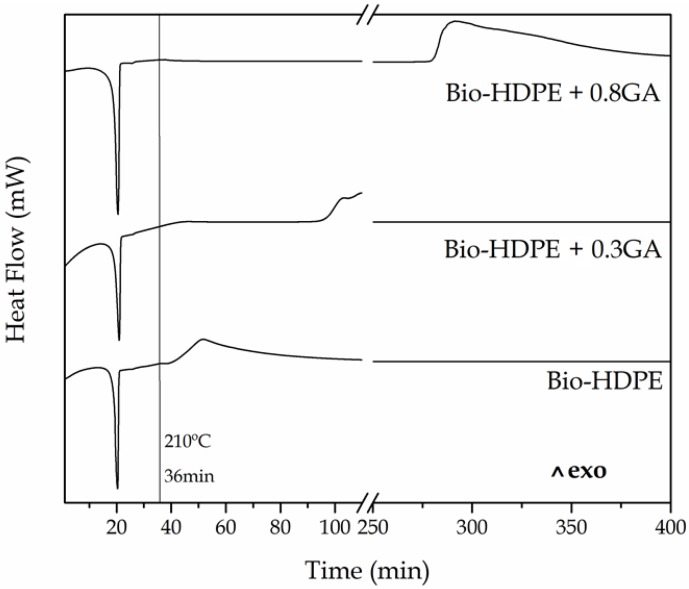
Differential scanning calorimetry (DSC) isothermal curves measured at 210 °C for a span time of 400 min of the bio-based high-density polyethylene (bio-HDPE) films containing different amounts of gallic acid (GA).

**Figure 4 polymers-12-00031-f004:**
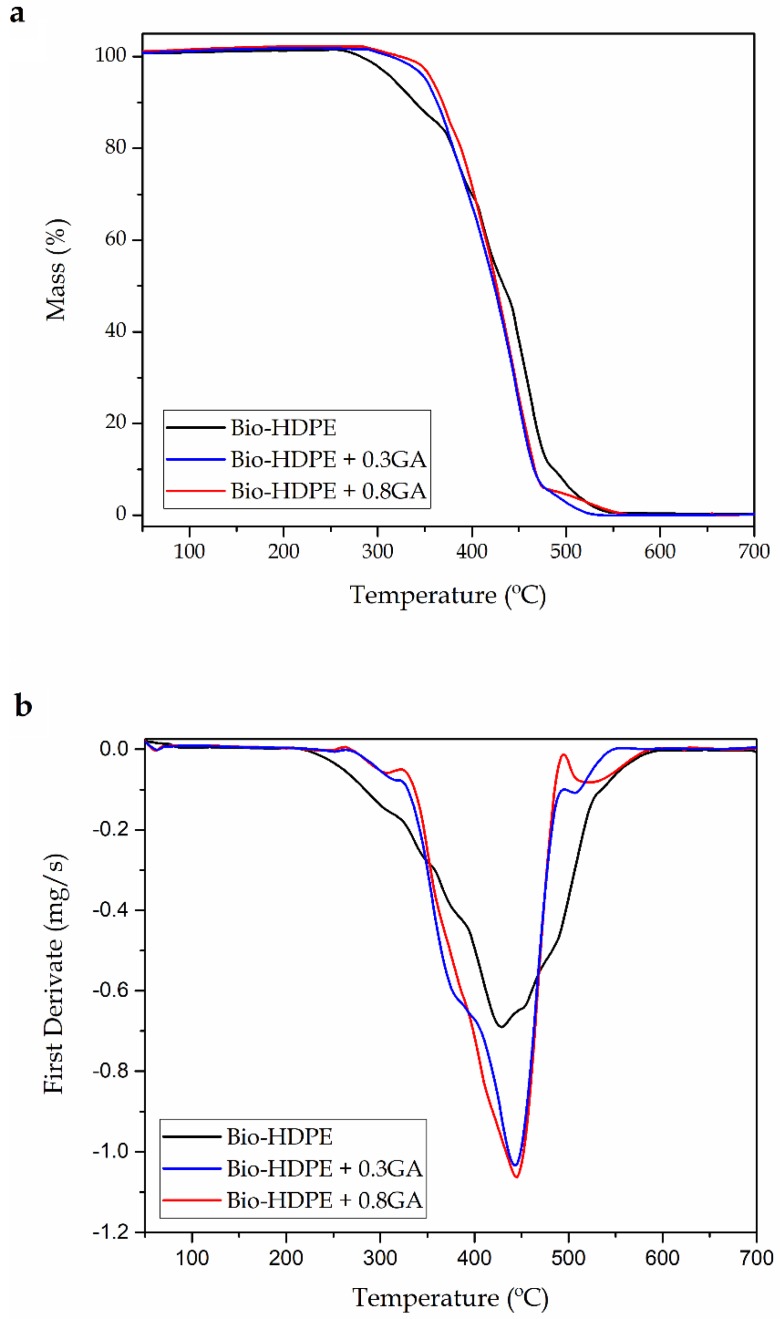
(**a**) Thermogravimetric analysis (TGA) curves and (**b**) first derivative (DTG) of the bio-based high-density polyethylene (bio-HDPE) films containing different amounts of gallic acid (GA).

**Figure 5 polymers-12-00031-f005:**
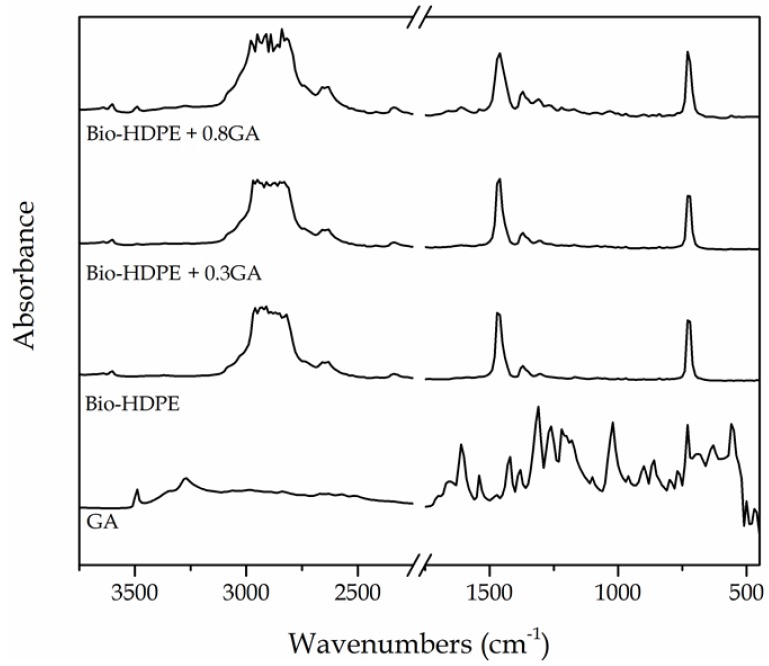
Fourier transform infrared (FTIR) spectra, from bottom to top, of gallic acid (GA) powder, bio-based high-density polyethylene (bio-HDPE) film, and bio-HDPE films containing 0.3 and 0.8 parts per hundred resin (phr) of GA.

**Figure 6 polymers-12-00031-f006:**
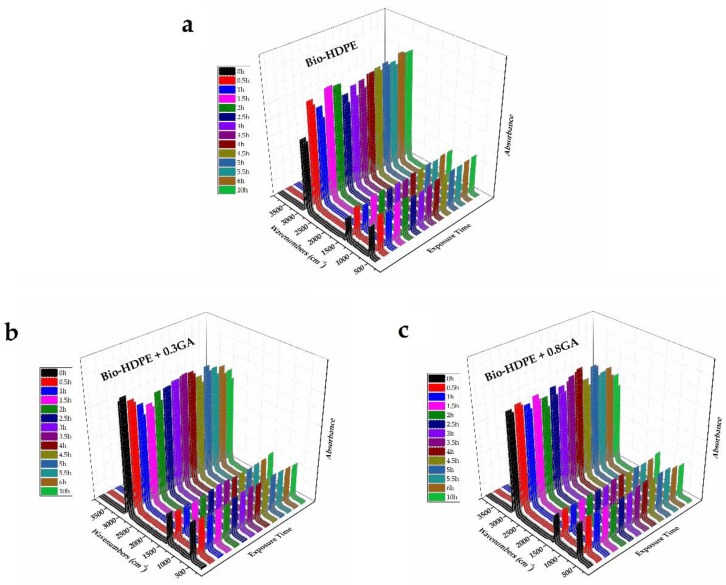
Fourier transform infrared (FTIR) spectra taken across the exposure time to ultraviolet (UV) light of the bio-based high-density polyethylene (bio-HDPE) films containing different amounts of gallic acid (GA): (**a**) Bio-HDPE; (**b**) Bio-HDPE + 0.3GA; (**c**) Bio-HDPE + 0.8GA.

**Figure 7 polymers-12-00031-f007:**
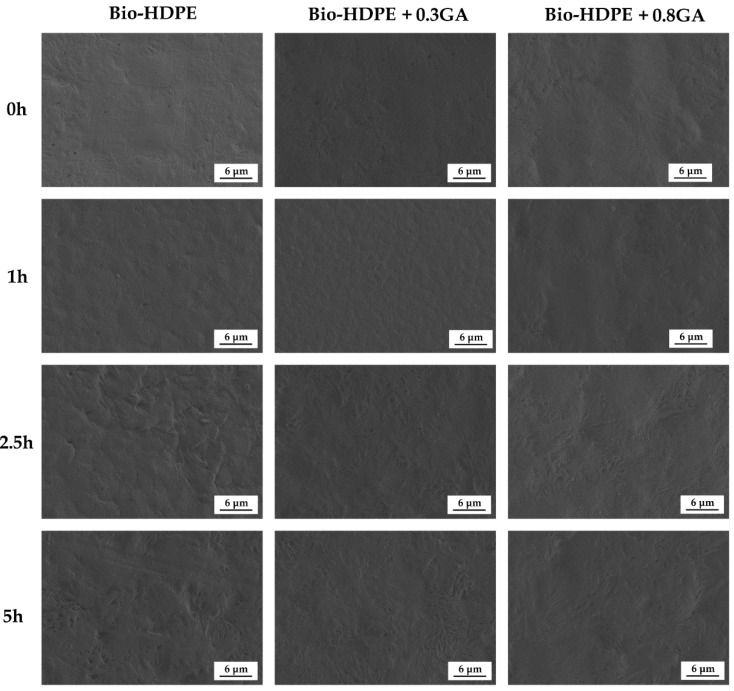
Field emission scanning electron microscopy (FESEM) micrographs of the bio-based high-density polyethylene (bio-HDPE) films containing different amounts of gallic acid (GA) exposed to different ultraviolet (UV) light exposure times.

**Table 1 polymers-12-00031-t001:** Color parameters (*L**, *a**, *b**, and Δ*E*_ab_*) of the bio-based high-density polyethylene (bio-HDPE) films containing different amounts of gallic acid (GA).

Film	*L**	*a**	*b**	Δ*E*_ab_***
Bio-HDPE	82.9 ± 1.0	−1.9 ± 0.1	−2.8 ± 0.3	-
Bio-HDPE + 0.3GA	75.3 ± 0.9	−0.7 ± 0.3	8.8 ± 0.4	13.9 ± 0.9
Bio-HDPE + 0.8GA	70.6 ± 0.5	−0.1 ± 0.1	10.9 ± 0.2	18.5 ± 0.4

**Table 2 polymers-12-00031-t002:** Tensile properties of the bio-based high-density polyethylene (bio-HDPE) films containing different amounts of gallic acid (GA) in terms of tensile modulus (*E*_tensile_), maximum tensile strength (σ_max_), and elongation at break (ε_b_).

Film	*E*_tensile_ (MPa)	σ_max_ (MPa)	ε_b_ (%)
Bio-HDPE	292.5 ± 22.1	21.3 ± 1.2	45.2 ± 3.5
Bio-HDPE + 0.3GA	222.1 ± 24.2	20.1 ± 0.6	18.6 ± 2.1
Bio-HDPE + 0.8GA	243.6 ± 31.5	20.8 ± 0.9	20.2 ± 2.3

**Table 3 polymers-12-00031-t003:** Thermal parameters of the bio-based high-density polyethylene (bio-HDPE) films containing different amounts of gallic acid (GA) in terms of melting temperature (*T*_m_), normalized melting enthalpy (Δ*H*_m_), degree of crystallinity (*X*_C_), onset oxidation temperature (OOT), and oxidation induction time (OIT).

Film	*T*_m_ (°C)	Δ*H*_m_ (J·g^−1^)	*X_C_* (%)	OTT (°C)	OIT (min)
Bio-HDPE	132.1 ± 0.3	160.6 ± 1.5	54.8± 0.8	226.3 ± 1.5	4.9 ± 0.3
Bio-HDPE + 0.3GA	132.4 ± 0.5	156.5 ± 1.4	53.4± 0.7	262.8 ± 2.1	60.8 ± 0.5
Bio-HDPE + 0.8GA	132.2 ± 0.7	152.7 ± 2.0	52.0 ± 0.9	270.2 ± 1.9	244.7 ± 1.0

**Table 4 polymers-12-00031-t004:** Thermal decomposition parameters of the bio-based high-density polyethylene (bio-HDPE) films containing different amounts of gallic acid (GA) in terms of onset degradation temperature (*T*_onset_), temperature of maximum degradation (*T*_deg_), and residual mass at 700 °C.

Film	*T*_onset_ (°C)	*T*_deg_ (°C)	Residual mass (%)
Bio-HDPE	256.9 ± 1.8	427.8 ± 1.3	0.22 ± 0.05
Bio-HDPE + 0.3GA	283.9 ± 2.0	442.9 ± 1.1	0.20 ± 0.04
Bio-HDPE + 0.8GA	291.6 ± 2.1	444.6 ± 1.2	0.17± 0.05

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
