# Peer review of "On the Use of Gallic Acid as a Potential Natural Antioxidant and Ultraviolet Light Stabilizer in Cast-Extruded Bio-Based High-Density Polyethylene Films"

_polymers, 2019, doi:10.3390/polym12010031_

Round 1
Reviewer 1 Report
The paper “On the use of Gallic Acid as a Natural Antioxidant and Ultraviolet Light Stabilizer in Cast-Extruded Bio-based High-density Polyethylene Films” reports on the study of Gallic acid as a natural antioxidant in bio-HDPE intended for food packaging applications. The paper is interesting and in my opinion the paper deserves to be published in Polymers after reviewing some points.
Introduction (Lines 56-57): please revise the typical processing conditions for polyethylenes, which are not around 220–240°C, but lower. Even in the reference [8] lower temperatures were used for processing HDPE. Mechanical properties: indeed, the decrease of crystallinity may cause the decrease of tensile modulus and strength but, in general, not a decrease of elongation. Please consider that a decrease of crystallinity means more amorphous polymer which has a higher mobility and therefore increased elongation. PLA is not the best example regarding the decrease of elongation (ref 34-36), because PLA is a brittle polymer but HDPE is much ductile and the decrease of elongation after GA addition is important. The dispersion of GA in the matrix could be a reason but the SEM images are blurry, too dark, and the details are not visible. I think the discussion regarding the decrease of elongation should be revised. TGA results: Please improve Figs 4a and b, they are not clear. Also, it is not very clear a single step decomposition in the case of bio-HDPE. The heating rate of 20 ° C.min-1 in the TGA could be too high. Why do you not choose 10 ° C.min-1? Do you have any explanation about the shoulder between 250 and 350 °C in the DTG curve of bio-HDPE? There are other references showing this early decomposition in HDPE? In my opinion, the UV aging tests do not show an improved ultraviolet (UV) resistance after GA addition in bio-HDPE. As mentioned in your work (lines 362-365) “Corrales et al. [66] showed that the UV degradation of HDPE generates an increase in the main peaks of acid groups (1712 cm-1), Aldehydes (1733 cm-1) Esters(1743 cm-1), Hydroxyls (3371 cm-1) and hydroperoxides free associate (3410 cm-1). However, none of these peaks was seen after aging. In these conditions, bio-HDPE seems to be stable to UV…If the role of GA to improve the UV resistance cannot be properly demonstrated, this must be removed even from the abstract. Please improve the quality of SEM images.
Author Response
The paper “On the use of Gallic Acid as a Natural Antioxidant and Ultraviolet Light Stabilizer in Cast-Extruded Bio-based High-density Polyethylene Films” reports on the study of Gallic acid as a natural antioxidant in bio-HDPE intended for food packaging applications. The paper is interesting and in my opinion the paper deserves to be published in Polymers after reviewing some points.
Q1) Introduction (Lines 56-57): please revise the typical processing conditions for polyethylenes, which are not around 220–240°C, but lower. Even in the reference [8] lower temperatures were used for processing HDPE.
A1) We have modified the range for the typical extrusion conditions and also included a new one showing actual temperatures in injection molding applications. Please see page 2, lines 54-56.
Q2) Mechanical properties: indeed, the decrease of crystallinity may cause the decrease of tensile modulus and strength but, in general, not a decrease of elongation. Please consider that a decrease of crystallinity means more amorphous polymer which has a higher mobility and therefore increased elongation. PLA is not the best example regarding the decrease of elongation (ref 34-36), because PLA is a brittle polymer but HDPE is much ductile and the decrease of elongation after GA addition is important. The dispersion of GA in the matrix could be a reason but the SEM images are blurry, too dark, and the details are not visible. I think the discussion regarding the decrease of elongation should be revised.
A2) The discussion about the effect of gallic acid on the mechanical properties of HDPE has been improved based on the reviewer comments and examples including HDPE studies have been added. Please see page 5 and 6, lines 216-222 and 228-234. We have also improved the resolution and size of the FESEM images.
Q3) TGA results: Please improve Figs 4a and b, they are not clear. Also, it is not very clear a single step decomposition in the case of bio-HDPE. The heating rate of 20 ° C.min-1 in the TGA could be too high. Why do you not choose 10 ° C.min-1? Do you have any explanation about the shoulder between 250 and 350 °C in the DTG curve of bio-HDPE? There are other references showing this early decomposition in HDPE?
A3) We have also improved the quality and size of this image. Please see new Figure 4. We have also performed the TGA tests at heating rates of 5 and 10 °C.min-1 and we found that the curves were very similar but the differences among the materials were smaller probably due to the higher stability of the samples at lower rates. For this reason, we kept the analysis at 20 °C.min-1. The presence of this shoulder was ascribed in the text to the early formation of free radicals, inducing an initial slower degradation rate, which was also suppressed in the GA-containing film samples. This was described in page lines 312-317 and 321-322 in pages 8 and 9.
Q4) In my opinion, the UV aging tests do not show an improved ultraviolet (UV) resistance after GA addition in bio-HDPE. As mentioned in your work (lines 362-365) “Corrales et al. [66] showed that the UV degradation of HDPE generates an increase in the main peaks of acid groups (1712 cm-1), Aldehydes (1733 cm-1) Esters(1743 cm-1), Hydroxyls (3371 cm-1) and hydroperoxides free associate (3410 cm-1). However, none of these peaks was seen after aging. In these conditions, bio-HDPE seems to be stable to UV…If the role of GA to improve the UV resistance cannot be properly demonstrated, this must be removed even from the abstract. Please improve the quality of SEM images.
A4) The reviewer is right that these peaks were not selected correctly to describe the UV degradation of the sample since their intensity was low and difficult to support by the current plots. We have rewritten this part based on the peak changes that can be observed in Figure 6.
Reviewer 2 Report
This manuscript describes the use of GA as UV light stabilizer in BioHDPE
The topic is interesting, but the manuscript should be improved.
I suggest publication of this paper after major revision. Please take into consideration the following comments and suggestions in the preparation of the revised manuscript:
Title – the antioxidant characteristics were not presented in the manuscript. Abstract- The units phr are not usual. May be use wt% indicating mass of Ga/mass of polymer.Please rewrite the last sentence. To which antioxidants are you referring?
In line 46 please explain why bioHDPE is called “microbial”.In line 72 please add more information explaining why GA was chosen.
Materials and methods.Table 1 – please change the units and delete the column of BioHDPE since it is constant.
Line 148- Should be Infrared .
Line 159- Scanning is missing
Results and DiscussionLine 186- Please explain in more detail in which foodstuff you are thinking.
Line 194- Mechanical properties should be discussed in more detail. Tensile modulus is the same as Elastic module? Why the elongation at break reduces with incorporation of GA and is so small compared to literature values?
Line 391- There is only this sentence about antioxidant effect of GA.
ConclusionsThe conclusions are too long. Please write only the main conclusions.
Author Response
This manuscript describes the use of GA as UV light stabilizer in BioHDPE
The topic is interesting, but the manuscript should be improved.
I suggest publication of this paper after major revision. Please take into consideration the following comments and suggestions in the preparation of the revised manuscript:
Q1) Title – the antioxidant characteristics were not presented in the manuscript. Abstract- The units phr are not usual. May be use wt% indicating mass of GA/mass of polymer.
A1) Gallic acid, GA, was used as an antioxidant based on its potential to delay thermal degradation. This is not related to the concept of active packaging (e.g. oxygen barrier by a reactive component in the film) but with the improvement of the thermal and UV light stability of the biopolymer in packaging films. This has been improved in the manuscript. Please see for instance page 12, lines 417-419. Alternatively, the phr units are widely used in the plastic industry for formulations in which some minor additives, which is the case of antioxidants and stabilizers, are incorporated with the polymers, fillers, etc.
Q2) Please rewrite the last sentence. To which antioxidants are you referring?
A2) The whole abstract has been rewritten to better explain the role of GA in the bio-HDPE films and also its contribution as a natural additive in the frame of the Bioeconomy.
Q3) In line 46 please explain why bioHDPE is called “microbial”.
A3) This was meant to explain that the monomer, ethylene, can also be obtained from biotechnological routes using microorganisms. Since we are not using this type of biopolymer, we have removed the word to avoid any missunderstanding.
Q4) In line 72 please add more information explaining why GA was chosen.
A4) We have improved the information included in the Introduction to better explain the reason to choose gallic acid. Please see page 2, lines 66-68 and 77-79.
Materials and methods.
Q5) Table 1 – please change the units and delete the column of BioHDPE since it is constant.
A5) The table was removed and its information was added in the text in section 2.2.
Q6) Line 148- Should be Infrared.
A6) This mistake was corrected.
Q7) Line 159- Scanning is missing
A7) The subsection title was changed to “Microscopy”
Results and Discussion
Q8) Line 186- Please explain in more detail in which foodstuff you are thinking.
A8) Some examples of food products were added in page 5, lines 188-196.
Q9) Line 194- Mechanical properties should be discussed in more detail. Tensile modulus is the same as Elastic module? Why the elongation at break reduces with incorporation of GA and is so small compared to literature values?
A9) Yes, tensile modulus and elastic modulus is the same. The effect of gallic acid on the mechanical properties of the bio-HDPE films has been better described in page 5, lines 216-222. The lower values of elongation at break can be ascribed to the testing conditions, manufacturing method, and differences in the percentage of crystallinity as well as crystals orientation. This has been indicated in lines 213-214. In any case, the objective of this test was to determine the effect of GA on the mechanical properties and not to compare the values with those reported in literature.
Q10) Line 391- There is only this sentence about antioxidant effect of GA.
A10) As indicated above, the concept of antioxidant was better described in the manuscript.
Conclusions
Q11) The conclusions are too long. Please write only the main conclusions.
A11) The conclusions were shortened.
Round 2
Reviewer 1 Report
I accept the paper in present form
Reviewer 2 Report
The authors addressed all the comments suggested by the reviewer and modified the manuscript accordingly. I think the manuscript can be accepted for publication.